# Embedded nano spin sensor for in situ probing of gas adsorption inside porous organic frameworks

Jie Zhang[1,2], Linshan Liu[1], Chaofeng Zheng[1,2], Wang Li[1], Chunru Wang [1] & Taishan Wang [1] ✉

Spin-based sensors have attracted considerable attention owing to their high sensitivities. Herein, we developed a metallofullerene-based nano spin sensor to probe gas adsorption within porous organic frameworks. For this, spin-active metallofullerene, $Sc_3C_2@C_{80}$, was selected and embedded into a nanopore of a pyrene-based covalent organic framework (Py-COF). Electron paramagnetic resonance (EPR) spectroscopy recorded the EPR signals of $Sc_3C_2@C_{80}$ within Py-COF after adsorbing $N_2$, CO, $CH_4$, $CO_2$, $C_3H_6$, and $C_3H_8$. Results indicated that the regularly changing EPR signals of embedded $Sc_3C_2@C_{80}$ were associated with the gas adsorption performance of Py-COF. In contrast to traditional adsorption isotherm measurements, this implantable nano spin sensor could probe gas adsorption and desorption with in situ, real-time monitoring. The proposed nano spin sensor was also employed to probe the gas adsorption performance of a metal–organic framework (MOF-177), demonstrating its versatility. The nano spin sensor is thus applicable for quantum sensing and precision measurements.

Recently, the use of electron spins for advanced science and technology has been actively pursued owing to their genuine quantum characteristics[1–4]. Consequently, spin-based sensors have gained significant attention owing to their high sensitivities and broad applications in probing physical and chemical parameters, such as electrical[5] or magnetic fields[3], molecule[6] or protein dynamics[7], and nuclei[8,9] or other particles. Along with this spin-based sensing research, quantum sensing has also emerged, causing a significant change in the paradigms of metrology. For example, spin-bearing diamonds with nitrogen-vacancy centers have been extensively studied for quantum sensing[10–13]. Moreover, several other spin systems could also potentially perform unique functions, such as quantum probing, which must be explored. Among these, molecular spins exhibit unique quantum properties, and certain proposals based on molecular spins for quantum computation and information storage have attracted considerable attention from physicists and chemists[14–16]. In particular, molecular spin systems feature numerous advantages, including the molecular-level chemical engineering of their spins. Moreover, molecular spins have independent nanoscale units that can realize in situ monitoring complex systems. Therefore, molecular spins demonstrate potential for quantum sensing applications.

Notably, paramagnetic endohedral metallofullerenes are nanoscale ball-like molecules with sensitive electron spins[17–20]. Owing to the presence of a fullerene cage, the endohedral electron spin presents high chemical stability. Moreover, the most significant advantages of such metallofullerene molecules are their susceptible electron spins, which can be potentially employed in quantum sensing applications. Accordingly, recent studies have explored the susceptible electron spins of metallofullerene molecules for sensing local magnetic fields, nanospace, molecule orientations, and molecule motions[20–24]. To promote these applications, it is, however, essential to explore appropriate sensing systems.

[1]Beijing National Laboratory for Molecular Sciences, Key Laboratory of Molecular Nanostructure and Nanotechnology, Institute of Chemistry, Chinese Academy of Sciences, Zhongguancun North First Street 2, Beijing 100190, China. [2]University of Chinese Academy of Sciences, Beijing 100049, China. ✉e-mail: wangtais@iccas.ac.cn

Porous organic frameworks are emerging crystalline porous materials with broad applications in gas adsorption and separation[25–28], gas storage[29,30], sensing[31–36], and catalysis[37–39]. Notably, covalent organic frameworks (COFs) and metal–organic frameworks (MOFs) have been investigated for extensive applications. For example, COFs have attracted significant interest owing to their wide surface areas and hierarchical nanopores, which are capable of encapsulating several guest species and causing chemical and physical changes. To reveal the adsorption mechanism and promote future applications, understanding and monitoring the nanopores within COFs, specifically detecting these nanopores in situ, are crucial steps. However, effective methods for such measurements are currently lacking. Notably, the aromatic fullerene shell of metallofullerenes endows them with lipophilicity, making them COF-nanopore-accessible. Therefore, paramagnetic endohedral metallofullerenes with sensitive electron spins can potentially sense COF nanopores in situ.

For a proof-of-concept study, we used a representative paramagnetic metallofullerene, $Sc_3C_2@C_{80}$, as an embedded spin probe to sense gas adsorption within a pyrene-based COF (Py-COF). Note that $Sc_3C_2@C_{80}$ has a highly stable electron spin protected by a carbon cage, and this stability forms the basis of its application. Moreover, $Sc_3C_2@C_{80}$ has a highly sensitive electron spin, as reported in our previous studies[18,40,41]. These features make $Sc_3C_2@C_{80}$ suitable for gas adsorption detection within porous materials. Py-COF, prepared using self-condensing building blocks with formyl and amino groups, is a recently emerging porous organic framework material with unique adsorption properties[42]. Its theoretical pore size is 1.38 nm; thus, one metallofullerene $Sc_3C_2@C_{80}$ unit (approximately 0.8 nm in size) can

enter one nanopore of Py-COF. Therefore, the molecular spin of $Sc_3C_2@C_{80}$ can effectively sense a nanopore of Py-COF in situ. In this study, the molecular spin of $Sc_3C_2@C_{80}$ was employed to probe the gas adsorption performance of Py-COF for $N_2$, CO, $CO_2$, $CH_4$, $C_3H_6$, and $C_3H_8$ gases by recording the electron paramagnetic resonance (EPR) signals of the embedded $Sc_3C_2@C_{80}$. The EPR intensities of the embedded $Sc_3C_2@C_{80}$ were observed to change when Py-COF adsorbed different gases. Therefore, the adsorption isotherms of Py-COF for different gases were recorded to comparatively illustrate the gas adsorption performance. Additionally, an in situ and real-time gas sensing process of Py-COF based on embedded $Sc_3C_2@C_{80}$ was designed, further demonstrating the application potential of this nano spin sensor. Moreover, the mechanism and evaluation methodology of this spin sensor were studied by analyzing the EPR signal intensity and linewidth. Finally, $Sc_3C_2@C_{80}$ was employed to probe the gas adsorption performance of an MOF (MOF-177), illustrating the versatility of this nano spin sensor.

## Results
### EPR studies of metallofullerene entrapped inside Py-COF
The basic concept underlying the metallofullerene-based nano spin sensor used for probing the gas adsorption performance of Py-COF is presented in Fig. 1a. Generally, when gas molecules are adsorbed onto COF or MOF materials, these molecules adhere to the pore frameworks and form gas layers on the frameworks[43–45]. These gas layers then restrict the motion of embedded $Sc_3C_2@C_{80}$, intensify spin–lattice interactions, and consequently weaken EPR signals. In this case, more gas molecules would enhance the thickness and viscosity of these gas

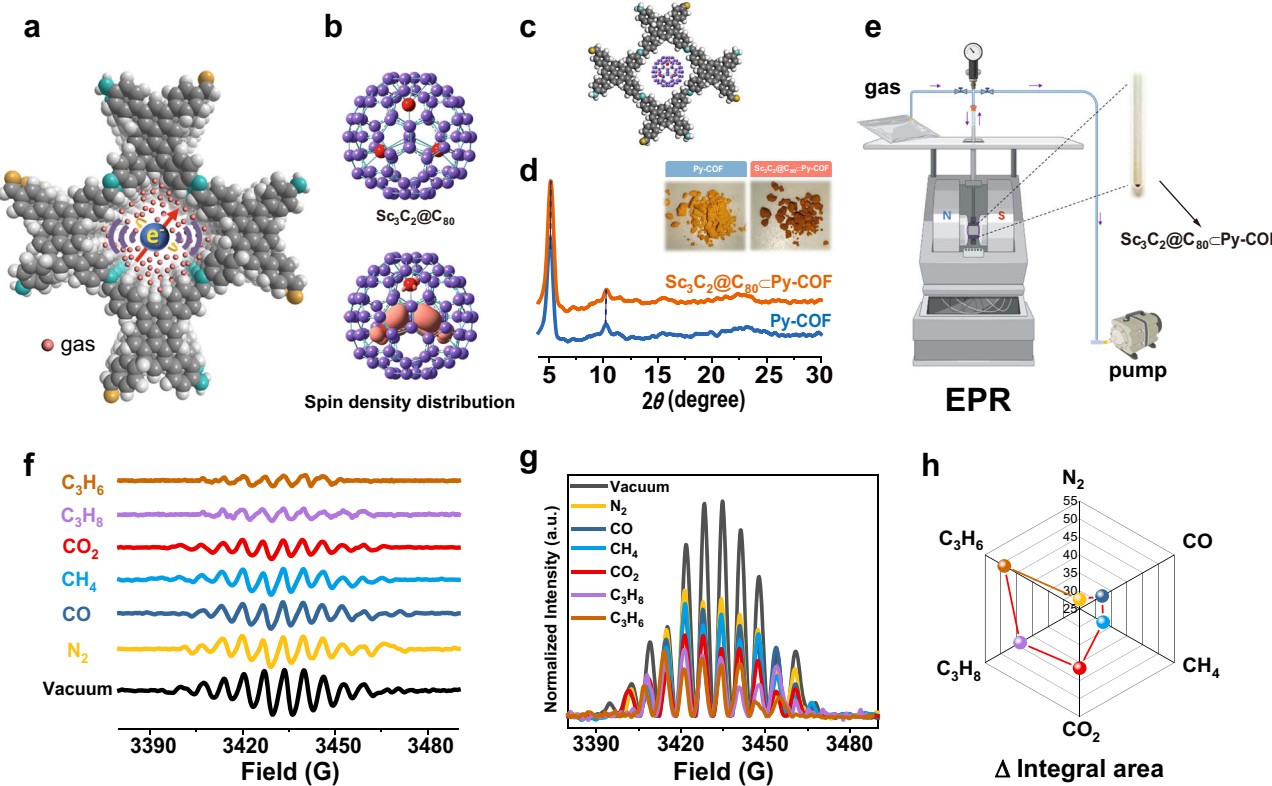

**Fig. 1 | Probing the gas adsorption performance of Py-COF using the molecular spin of $Sc_3C_2@C_{80}$. a** Schematic of the nano spin sensor in a Py-COF channel. **b** Structure and spin density distributions of $Sc_3C_2@C_{80}$. **c** Theoretically computed unit structure of $Sc_3C_2@C_{80}⊂Py$-COF. **d** Powder X-ray diffraction (PXRD) patterns of Py-COF (Blue) and $Sc_3C_2@C_{80}⊂Py$-COF (Orange). The insets present the sample pictures of Py-COF and $Sc_3C_2@C_{80}⊂Py$-COF. **e** Sketch of the experimental setup, which allows vacuuming, inflation, and EPR measurements of $Sc_3C_2@C_{80}⊂Py$-COF

in a quartz tube. Figure created with BioRender.com. **f** EPR spectra of $Sc_3C_2@C_{80}⊂Py$-COF under vacuum and after $N_2$, CO, $CH_4$, $CO_2$, $C_3H_8$, and $C_3H_6$ adsorption. **g** Integrated EPR spectra of $Sc_3C_2@C_{80}⊂Py$-COF under different conditions. **h** Difference values between the integrated EPR signal area of $Sc_3C_2@C_{80}⊂Py$-COF after adsorbing one gas and the counterpart of $Sc_3C_2@C_{80}⊂Py$-COF under vacuum conditions. Source data are provided as a Source Data file.

layers and further weaken the EPR signals of embedded $Sc_3C_2@C_{80}$. For this testing platform, a nano spin sensor involving $Sc_3C_2@C_{80}$ embedded in a Py-COF channel can sense gas adsorption through spin–lattice interactions, which influence the EPR signal intensity and linewidth.

Typically, Py-COF is synthesized using bifunctional monomeric 1,6-bis(4-formylphenyl)–3,8-bis(4-aminophenyl)pyrene, with two formyl and two amino groups, through intermolecular self-condensation, forming a 2D pyrene-based imine COF[42]. Py-COF demonstrates good thermal stability, as revealed by thermogravimetric analysis (Supplementary Fig. 1). In our study, $Sc_3C_2@C_{80}$ was synthesized using the Kräschmer−Huffman arc-discharge method[46], and Supplementary Fig. 2 shows the preparation and characterization details. As presented in Fig. 1b, $Sc_3C_2@C_{80}$ has an unpaired electron localized in the $Sc_3C_2$ cluster. A theoretically computed unit structure of $Sc_3C_2@C_{80} \subset$ Py-COF is shown in Fig. 1c. The theoretical pore size of Py-COF is 1.38 nm; thus, a pore of Py-COF can effectively accommodate a metallofullerene $Sc_3C_2@C_{80}$ unit (approximately 0.8 nm in size) through host−guest interactions, including van der Waals forces and certain π−π interactions. The predicted position of $Sc_3C_2@C_{80}$ within Py-COF pores indicates that the molecular spin of $Sc_3C_2@C_{80}$ can sense the environment of a Py-COF nanopore in situ. In addition, the encapsulation of $Sc_3C_2@C_{80}$ inside the channel of a Py-COF would not substantially prevent gases from infiltrating, and ample space may be available for gas adsorption, as observed from Fig. 1c.

$Sc_3C_2@C_{80}$ was entrapped in Py-COF using the adsorption method in a toluene solution, yielding a complex of $Sc_3C_2@C_{80} \subset$ Py-COF, as shown in Fig. 1d and Supplementary Fig. 3a. After the adsorption of $Sc_3C_2@C_{80}$, the color of Py-COF changed from light yellow to brown, and the obtained complex was sufficiently washed using toluene to remove $Sc_3C_2@C_{80}$ physically adhered on the Py-COF surface. Moreover, the mass fraction of $Sc_3C_2@C_{80}$ in Py-COF was calculated to be 1.5% (Supplementary Fig. 3b), and the analysis process is presented in Supplementary Note 1. Next, PXRD experiments were performed to characterize the crystallinity of Py-COF and $Sc_3C_2@C_{80} \subset$ Py-COF, as shown in Fig. 1d. A comparison between their PXRD patterns indicated that $Sc_3C_2@C_{80}$ did not disrupt the crystal structure of Py-COF. The detection of the Sc $2p$ peak in $Sc_3C_2@C_{80} \subset$ Py-COF via X-ray photoelectron spectroscopy (XPS) indicated that $Sc_3C_2@C_{80}$ molecules were trapped inside the channels of Py-COF, as shown in Supplementary Fig. 4. Scanning electron microscopy (SEM) images of Py-COF and $Sc_3C_2@C_{80} \subset$ Py-COF revealed that the composite material morphology and size did not change after preparation (Supplementary Fig. 5). The transmission electron microscopy (TEM) element mapping images and energy dispersive spectroscopy results revealed that $Sc_3C_2@C_{80}$ was uniformly distributed inside Py-COF (Supplementary Fig. 6).

The EPR spectra of $Sc_3C_2@C_{80} \subset$ Py-COF under vacuum and after adsorbing the gases are shown in Fig. 1f. The apparent distinct EPR splitting peaks indicate successful embedment and dispersion of $Sc_3C_2@C_{80}$ molecules, as simple physical dispersion could lead to the aggregation of $Sc_3C_2@C_{80}$, which would present a broad single EPR peak. In addition, the EPR spectrum of $Sc_3C_2@C_{80}$ in a $CS_2$ solution presents 22 splitting peaks owing to the couplings between spin and three equivalent Sc nuclei, as shown in Supplementary Fig. 13. However, the EPR spectrum of $Sc_3C_2@C_{80} \subset$ Py-COF under vacuum presented broadened EPR signals owing to the strong host−guest interaction between $Sc_3C_2@C_{80}$ and the pores of Py-COF, which can intensify spin–lattice interactions and change the EPR signals.

Subsequently, $Sc_3C_2@C_{80}$ was employed to probe the gas adsorption performance of Py-COF for $N_2$, CO, $CO_2$, $CH_4$, $C_3H_6$, and $C_3H_8$ by recording the EPR signals. Figure 1e presents a sketch of the experimental setup. The EPR spectra of $Sc_3C_2@C_{80} \subset$ Py-COF varied under different conditions. Briefly, after adsorbing the above gases, the EPR signal intensity of $Sc_3C_2@C_{80} \subset$ Py-COF decreased compared to

that under vacuum conditions, and the EPR signals of $Sc_3C_2@C_{80} \subset$ Py-COF changed.

To further analyze the EPR signals under different conditions, integral EPR spectra were obtained, as shown in Fig. 1g. The intensities of the EPR signals under vacuum conditions were the strongest, and the intensities under gas adsorption conditions gradually decreased in the following order: $N_2$, CO, $CH_4$, $CO_2$, $C_3H_8$, and $C_3H_6$. In addition, the difference values between the integrated EPR signal intensity (integrated EPR signal area) of $Sc_3C_2@C_{80} \subset$ Py-COF before and after gas adsorption are plotted in Fig. 1h, which demonstrates that the difference values of the integrated EPR signal intensity gradually increased in the following order: $N_2$, CO, $CH_4$, $CO_2$, $C_3H_8$, and $C_3H_6$.

As discussed above, when gas molecules adhere to porous frameworks, gas layers are formed on them. These gas layers restrict the motion of embedded $Sc_3C_2@C_{80}$ and intensify spin–lattice interactions. In this scenario, more gas molecules would enhance the viscosity of these gas layers and influence the rotational relaxation time of $Sc_3C_2@C_{80}$ molecules. The relationship between the viscosity $\eta$ and rotational correlation time $\tau_c$ in the spin system can be expressed as[47]

$$\tau_c = \frac{4\pi r^3 \eta}{3kT} \quad (1)$$

where $r$ is the dynamic radius of a spin molecule. The spin–lattice relaxation time ($T_1$) is expressed as follows[48]:

$$\frac{1}{T_1} = \frac{2\pi g \beta_e}{h}\left(B_x^2 + B_y^2\right)\frac{\tau_c}{1 + \tau_c^2 \omega_s^2} \quad (2)$$

where $B_x^2 + B_y^2$ denote the square amplitudes of fluctuating fields along the x- and y-directions. According to Eqs. (1) and (2), a larger viscosity could result in a smaller $T_1$. Furthermore, the double-integrated intensity (DIN) can be expressed by the Arrhenius equation[49]:

$$DIN(T) = I_0 \exp\left(\frac{E_a}{kT}\right) \quad (3)$$

where $I_0$ is a fitting parameter, and $E_a$ is the activation energy required for dissociation of the paramagnetic spin cluster. Combining $T_1$ with Eq. (3) $1/T_1 \propto \exp(-E_a/kT)$[50] reveals that the integrated EPR intensity would decrease with a decrease in $T_1$. Therefore, a corresponding relationship exists between gas adsorption and EPR signals.

## Gas adsorption analysis of Py-COF
Next, the adsorption capacities of Py-COF for the aforementioned gases were then measured based on adsorption isotherms, as shown in Fig. 2a and Supplementary Fig. 7. Py-COF demonstrated increasing adsorption capacity at 1 bar for the above gases in the following order: $N_2$, CO, $CH_4$, $CO_2$, $C_3H_8$, and $C_3H_6$. Notably, this order of adsorption capacity was identical to that of the changed EPR signals of $Sc_3C_2@C_{80} \subset$ Py-COF before and after adsorbing different gases at 1 bar, as shown in Fig. 2b. Thus, in addition to the increase in the adsorption amount of Py-COF, the difference between the integrated EPR signal intensity for $Sc_3C_2@C_{80} \subset$ Py-COF before and after adsorbing gas increased. This unique relationship indicates that embedded $Sc_3C_2@C_{80}$ can be used as a molecular nanosensor to probe the gas adsorption performance of Py-COF and other porous organic frameworks.

In addition, differences between the change trends obtained based on the adsorption isotherm and EPR measurements were observed. This was because the commonly used adsorption isotherm test analyzes the gas adsorption of all pores within Py-COF, whereas EPR measurements only reflect the gas adsorption near $Sc_3C_2@C_{80}$ in a nanoscale channel of Py-COF. As shown in Supplementary Fig. 10, Py-COF displays a combined type I dominant isotherm with a sharp

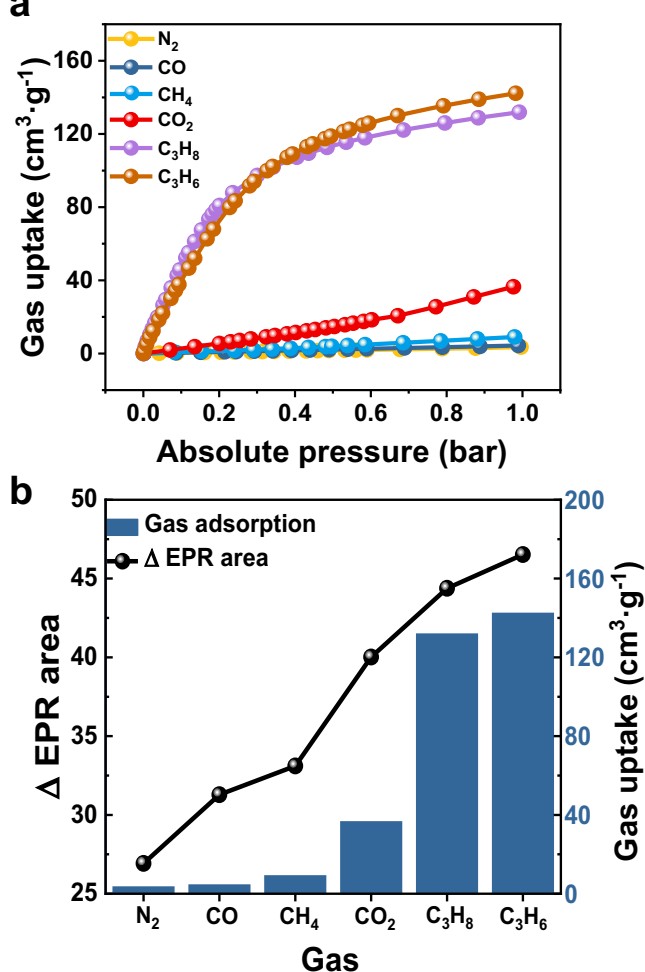

**Fig. 2 | Relationship between gas adsorption performance and EPR signals. a** Recorded adsorption isotherms of Py-COF for $N_2$, CO, $CH_4$, $CO_2$, $C_3H_8$, and $C_3H_6$. **b** Measured adsorption amounts of different gases for Py-COF using adsorption isotherms at 1 bar (histogram) and changed difference values for integrated EPR signal areas of $Sc_3C_2@C_{80}\subset$Py-COF before and after adsorbing different gases at 1 bar (point plot). Source data are provided as a Source Data file.

uptake in the low relative pressure region[51] ($P/P_0 < 0.1$), a characteristic of microporous materials. Although the pore size of Py-COF demonstrated a theoretical distribution centered at 1.38 nm, some sub-nanoscale cracks could be observed. These cracks could result in a non-equivalent relationship between the amounts obtained from the adsorption isotherm and EPR measurements. Therefore, the $N_2$ sorption analysis revealed the average porosity of Py-COF. However, EPR measurements with a spin sensor involving $Sc_3C_2@C_{80}$ entrapped in Py-COF could be used to probe the gas adsorption performance and adsorption and desorption processes with in situ and real-time monitoring.

In addition, we performed contrast adsorption isotherm experiments on $C_3H_6$ and $C_3H_8$ gases for Py-COF and $C_{60}\subset$Py-COF (mass fraction: 9.8%), indicating that the adsorption capacity of Py-COF did not change after fullerene encapsulation (Supplementary Fig. 8). Thus, a small amount of fullerene inside Py-COF would not prevent the gas from infiltrating.

### Probing gas adsorption of COFs with a nano spin sensor
Generally, sensitivity is an essential parameter for a probe. For the proposed nano spin sensor, detecting gas adsorption at low pressures is a

significant indicator of sensitivity. To investigate the sensitivity of this spin probe, we recorded EPR signals at 0.2 bar and 0.6 bar for $N_2$, CO, $CH_4$, and $CO_2$, as shown in Fig. 3a, b. After adsorption of $N_2$, CO, $CH_4$, and $CO_2$ at 0.2 bar and 0.6 bar, the EPR signal intensity of $Sc_3C_2@C_{80}\subset$Py-COF decreased compared to that under vacuum conditions. Moreover, integral EPR spectra were obtained, and the intensities under gas adsorption conditions of 0.2 bar and 0.6 bar decreased in the following order: $N_2$, CO, $CH_4$, and $CO_2$. The trends of the changed difference values for the integrated EPR signal areas of $Sc_3C_2@C_{80}\subset$Py-COF before and after adsorbing different gases at 0.2 bar and 0.6 bar are shown in Fig. 3c, and the difference values for the integrated EPR signal areas appear to be significant under low pressures. In addition, the adsorption capacity of Py-COF for the gases ($N_2$, CO, $CH_4$, and $CO_2$) could still be well distinguished at 0.2 bar and 0.6 bar (Supplementary Fig. 11).

Moreover, the EPR signals after gas adsorption under lower pressures were recorded, as shown in Supplementary Fig. 9. These EPR tests were performed under the following conditions: 0.12 bar $CO_2$, 0.03 bar $C_3H_8$, and 0.04 bar $C_3H_6$ see in Supplementary Note 3. $Sc_3C_2@C_{80}\subset$Py-COF was able to sense gas adsorption at extremely low pressures, revealing the high sensitivity of this molecular spin probe based on metallofullerene $Sc_3C_2@C_{80}$.

To date, relevant EPR methods to probe gas adsorption have seldom been reported; yet there are few reports on the EPR properties of TEMPO radicals within MOF materials. For example, for a TEMPO radical and ZIF-8 complex, the adsorption of different gases at atmospheric pressure was found to change the rotational correlation time of the TEMPO radical[52]. In addition, the TEMPO radical and ZIF-8 complex were involved in investigations on high-pressure induced amorphization[53]. We consider this molecular spin probe based on metallofullerene $Sc_3C_2@C_{80}$ for the gas adsorption measurement of porous organic frameworks to be closer to sensing application.

As mentioned, EPR measurements using a nano spin sensor involving $Sc_3C_2@C_{80}$ entrapped in a Py-COF could be used to probe the adsorption and desorption processes. To illustrate this concept, we designed a group of EPR experiments in which $C_3H_6$ gas was selected as a detection object. As shown in Fig. 4a, four states of $Sc_3C_2@C_{80}\subset$Py-COF were selected to record the EPR spectra, with 20 repeated measurements performed for each state to avoid instrumental errors. $Sc_3C_2@C_{80}\subset$Py-COF under vacuum, State I, presented high-intensity integrated EPR signals (Fig. 4b and Supplementary Fig. 12). However, after absorbing $N_2$, State II, the integrated EPR signals of $Sc_3C_2@C_{80}\subset$Py-COF decreased slightly in intensity. Notably, after absorbing $C_3H_6$, State III, the integrated EPR signals of $Sc_3C_2@C_{80}\subset$Py-COF decreased significantly in intensity. Finally, after another vacuum treatment, State IV, $Sc_3C_2@C_{80}\subset$Py-COF could recover its high-intensity EPR signals. These results demonstrate that the nano spin sensor involving $Sc_3C_2@C_{80}$ entrapped in a Py-COF could detect the existence or leakage of $C_3H_6$ gas, an essential chemical material.

In addition, we compared the EPR signals of different gases at the same adsorption capacity and discovered that the integrated EPR intensity variations were similar, as shown in Supplementary Fig. 9. This indicates that the EPR signals of $Sc_3C_2@C_{80}$ entrapped within porous materials mainly depend on the gas amounts around the spin probe. Therefore, this nano spin sensor is suitable for probing and evaluating the gas adsorption performance of porous materials.

Further, EPR signal linewidths were analyzed for the nano spin sensor involving $Sc_3C_2@C_{80}$ entrapped in Py-COF to reveal the probing mechanism. Notably, compared with the 22 hyperfine coupling splitting peaks of $Sc_3C_2@C_{80}$ in a solution, the free rotation of $Sc_3C_2@C_{80}$ embedded in Py-COF is restricted owing to strong host–guest interactions, resulting in significant linewidth broadening, as shown in Fig. 1f. According to Kivelson's theory[54], the linewidth ($\Delta H$) of EPR spectra depends on the quantum number ($M_I$) and can be

 

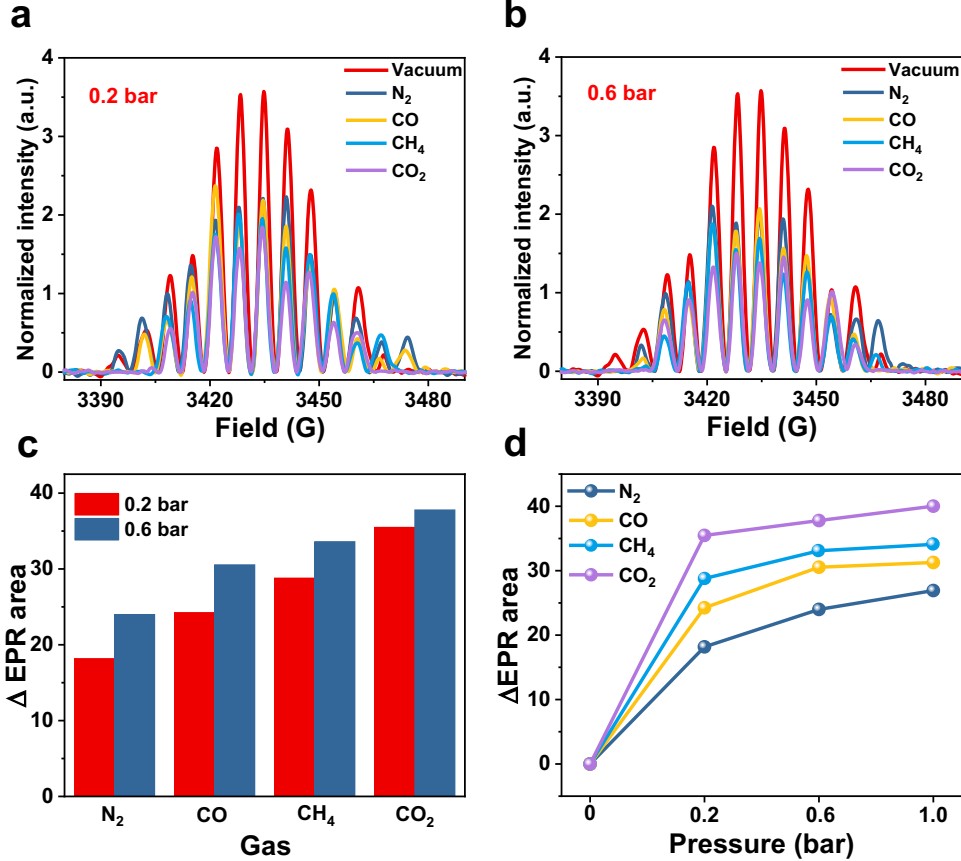

**Fig. 3 | Effect of gas pressure on EPR signals.** Integrated EPR spectra of $Sc_3C_2@C_{80}⊂Py$-COF under vacuum and after adsorption of $N_2$, CO, $CH_4$, and $CO_2$ at **a** 0.2 bar and **b** 0.6 bar. **c** Difference values of integrated EPR signal areas for $Sc_3C_2@C_{80}⊂Py$-COF before and after adsorbing $N_2$, CO, $CH_4$, and $CO_2$ at 0.2 bar and 0.6 bar. **d** Trends of the changed difference values for the integrated EPR signal areas of $Sc_3C_2@C_{80}⊂Py$-COF before and after adsorbing different gases at 0.2, 0.6, and 1 bar. Source data are provided as a Source Data file.

expressed as

$$\triangle H = \alpha + \beta M_I + \gamma M_I^2 + \delta M_I^4 \qquad (4)$$

where the $\alpha$, $\beta$, $\gamma$, and $\delta$ parameters depend on the rotational correlation time $\tau_c$[47]. From Eq. (4), it can be inferred that the linewidth was enlarged at both the wing parts of the EPR spectra, and at certain times, a high value of $M_I$ could cause partial disappearance of the side EPR signals[55]. Different EPR signal linewidths of $Sc_3C_2@C_{80}⊂Py$-COF under vacuum and after adsorbing $N_2$ and $C_3H_6$ at 1 bar successively are shown in Fig. 5. After adsorbing $N_2$, $Sc_3C_2@C_{80}⊂Py$-COF demonstrated slightly increased linewidth compared with $Sc_3C_2@C_{80}⊂Py$-COF under vacuum conditions, revealing that adsorbed $N_2$ could further restrict the rotation of embedded $Sc_3C_2@C_{80}$. Furthermore, after adsorbing $C_3H_6$ successively, $Sc_3C_2@C_{80}⊂Py$-COF had a significantly increased linewidth, revealing the severely restricted rotation of embedded $Sc_3C_2@C_{80}$ after adsorbing $C_3H_6$. Thus, adsorbed gases in Py-COF can result in a longer rotational correlation time $\tau_c$ and a larger linewidth for the molecular spin probe involving $Sc_3C_2@C_{80}$.

## EPR studies of metallofullerene entrapped inside MOF-177

To explore the versatility of this metallofullerene-based nano spin sensor, $Sc_3C_2@C_{80}$ was employed to probe the gas adsorption performance of a MOF (MOF-177), as shown in Fig. 6. For this, $Sc_3C_2@C_{80}$ was entrapped in MOF-177 using an adsorption method in a toluene solution, yielding a complex of $Sc_3C_2@C_{80}⊂$MOF-177. The mass fraction of $Sc_3C_2@C_{80}$ in MOF-177 was calculated to be 3.78%, and the analysis process is presented in Supplementary Note 4 and

Supplementary Fig. 14. The MOF-177 and $Sc_3C_2@C_{80}⊂$MOF-177 complexes were characterized using SEM, which revealed that their morphologies were similar (Supplementary Fig. 15). Moreover, TEM element mapping indicated that $Sc_3C_2@C_{80}$ molecules were dispersed in the MOF-177 (Supplementary Fig. 16).

Subsequently, EPR measurements were employed to probe the gas adsorption performance of MOF-177 for $N_2$, $H_2$, $CO_2$, $CH_4$, $C_3H_6$, and $C_3H_8$. The EPR spectra of $Sc_3C_2@C_{80}⊂$MOF-177 under vacuum and after adsorbing the foregoing gases are shown in Supplementary Fig. 16a, and the corresponding integral EPR spectra are depicted in Fig. 6b. After adsorbing these gases, the EPR signal intensity of $Sc_3C_2@C_{80}⊂$MOF-177 decreased compared to that under vacuum conditions. In addition, the difference values between the integrated EPR signal intensity of $Sc_3C_2@C_{80}⊂$MOF-177 before and after adsorbing gases are illustrated in Fig. 6c, demonstrating the following increasing order for the difference values of the integrated EPR signal intensity: $N_2$, $H_2$, $CH_4$, $CO_2$, $C_3H_8$, and $C_3H_6$.

The adsorption capacity of MOF-177 for the same gases was then measured based on adsorption isotherms, as shown in Supplementary Fig. 16b. Notably, MOF-177 presents increasing adsorption capacity at 1 bar for the gases in the following order: $N_2$, $H_2$, $CH_4$, $CO_2$, $C_3H_8$, and $C_3H_6$. As shown in Fig. 6d and Supplementary Fig. 17, a corresponding relationship exists between the measured adsorption amounts of MOF-177 and changed EPR signals of $Sc_3C_2@C_{80}⊂$MOF-177 before and after adsorbing different gases at 1 bar. This distinct relationship reveals that $Sc_3C_2@C_{80}$ embedded in MOF-177 can be employed as a nano spin sensor to probe the gas adsorption performance of MOF-177. In addition, differences still persist between the change trends

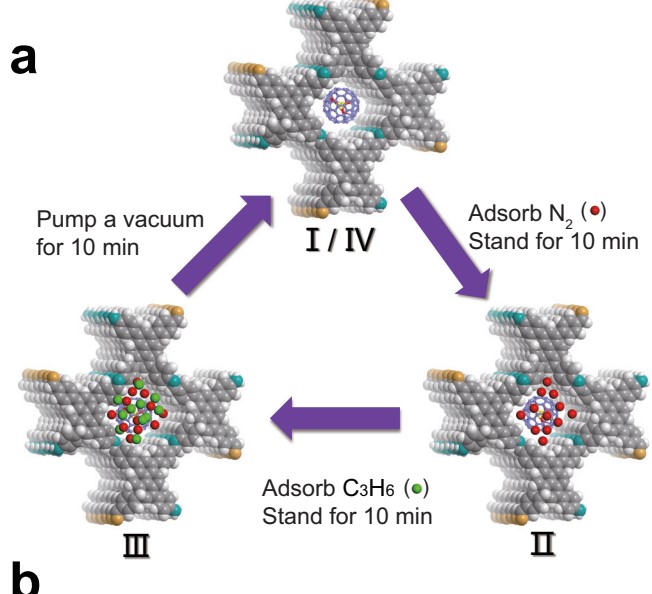

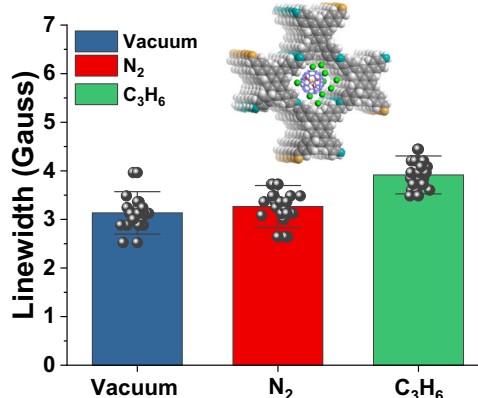

**Fig. 5 | Analysis of EPR signal linewidth.** Different EPR signal linewidths of $Sc_3C_2@C_{80} \subset$ Py-COF under vacuum and after successive adsorption of $N_2$ and $C_3H_6$ at 1 bar. Twenty EPR spectra were collected for each state. Error bar in graphs represent the ±Standard Deviation (SD). Source data are provided as a Source Data file.

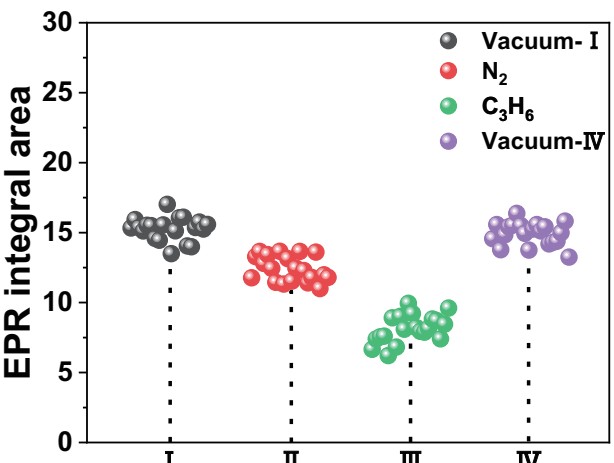

**Fig. 4 | Probing the gas adsorption process of Py-COF with the molecular spin of $Sc_3C_2@C_{80}$. a** Schematic of the gas adsorption/desorption processes of $Sc_3C_2@C_{80} \subset$ Py-COF. States I denotes $Sc_3C_2@C_{80} \subset$ Py-COF under vacuum. States II and III denote $Sc_3C_2@C_{80} \subset$ Py-COF after absorbing $N_2$ and $C_3H_6$ at 1 bar successively. State IV denotes $Sc_3C_2@C_{80} \subset$ Py-COF under a vacuum state. **b** Recorded integrated EPR signal areas for the above four states. Twenty EPR spectra were collected for each state. Source data are provided as a Source Data file.

obtained from the adsorption isotherm and EPR measurements because the EPR measurement using a spin sensor involving $Sc_3C_2@C_{80}$ entrapped in MOF-177 is capable of probing gas with in situ and real-time monitoring. These results demonstrate that $Sc_3C_2@C_{80}$ embedded in porous organic frameworks can act as a nano spin sensor to probe the gas adsorption performance and adsorption process.

## Discussion

$Sc_3C_2@C_{80}$ was embedded into the pores of a Py-COF, and its molecular spin was employed to probe the gas adsorption performance of Py-COF for $N_2$, CO, $CO_2$, $CH_4$, $C_3H_6$, and $C_3H_8$. When combined with traditional adsorption isotherm measurements, $Sc_3C_2@C_{80}$ exhibited considerably decreased EPR signals upon gas adsorption. The adsorbed gases in Py-COF could restrict the rotation of embedded $Sc_3C_2@C_{80}$ and result in a longer rotational correlation time, a larger linewidth, and decreased intensity for the molecular $Sc_3C_2@C_{80}$ nanosensor. We investigated the sensitivity of this spin probe by

recording EPR signals after adsorbing $N_2$, CO, $CH_4$, and $CO_2$ at 0.2 bar and 0.6 bar; the difference values for the integrated EPR signal intensities were significant, revealing the high sensitivity of the spin probe at low pressures. This high sensitivity was further verified by testing $CO_2$, $C_3H_8$, and $C_3H_6$ at 0.12, 0.03, and 0.04 bar, respectively, where these gases at lower pressures were similarly detected. Furthermore, the proposed $Sc_3C_2@C_{80} \subset$ Py-COF spin sensor was used to probe the adsorption and desorption processes of $C_3H_6$, which revealed that $Sc_3C_2@C_{80} \subset$ Py-COF could be used to detect the leakage of $C_3H_6$ gas, an important and hazardous chemical.

The versatility of this $Sc_3C_2@C_{80}$ spin sensor was explored by probing the gas adsorption performance of MOF-177 for $N_2$, $H_2$, $CO_2$, $CH_4$, $C_3H_6$, and $C_3H_8$. These results demonstrated that $Sc_3C_2@C_{80}$ embedded in other porous organic frameworks can also act as a nano spin sensor to probe the gas adsorption performance, gas adsorption and desorption processes, and gas leakage.

The changed difference values for the integrated EPR signal areas (ΔEPR) of $Sc_3C_2@C_{80}$ entrapped within porous materials before and after gas adsorption are dependent on the amount of entrapped $Sc_3C_2@C_{80}$, as well as the interaction between $Sc_3C_2@C_{80}$ and nanopores. A larger amount of embedded $Sc_3C_2@C_{80}$ would result in a larger ΔEPR; however, the interactions between $Sc_3C_2@C_{80}$ and nanopores would influence the EPR signal and ΔEPR. Therefore, a comparative analysis of ΔEPR could be employed for one type of complex. For example, the amount of $Sc_3C_2@C_{80}$ in Py-COF is low (mass fraction: 1.5%); hence, the integrated EPR signal area and ΔEPR are relatively small compared to those of $Sc_3C_2@C_{80} \subset$ MOF-177, which has a higher amount of $Sc_3C_2@C_{80}$ in MOF-177 (mass fraction: 3.78%). In addition, the different interactions between $Sc_3C_2@C_{80}$ and nanopores of Py-COF and MOF-177 would influence the initial EPR signals. Therefore, when the proposed nano spin sensor is employed to probe the gas adsorption performance, ΔEPR after adsorbing one gas should be compared with the counterpart under vacuum conditions for one type of complex.

Notably, the spin sensor involving $Sc_3C_2@C_{80}$ embedded in porous organic frameworks can probe gas adsorption and desorption processes with in situ and real-time monitoring. Traditional adsorption isotherm measurements often indicate the average porosity of porous organic frameworks. Therefore, differences are often noted between the change trends obtained from adsorption isotherms and EPR measurements. Compared with the adsorption amounts measured using adsorption isotherms, the EPR signals have more significant difference values for each gas adsorption (Figs. 2a and 6d). The reason for this is that the commonly used adsorption isotherm method

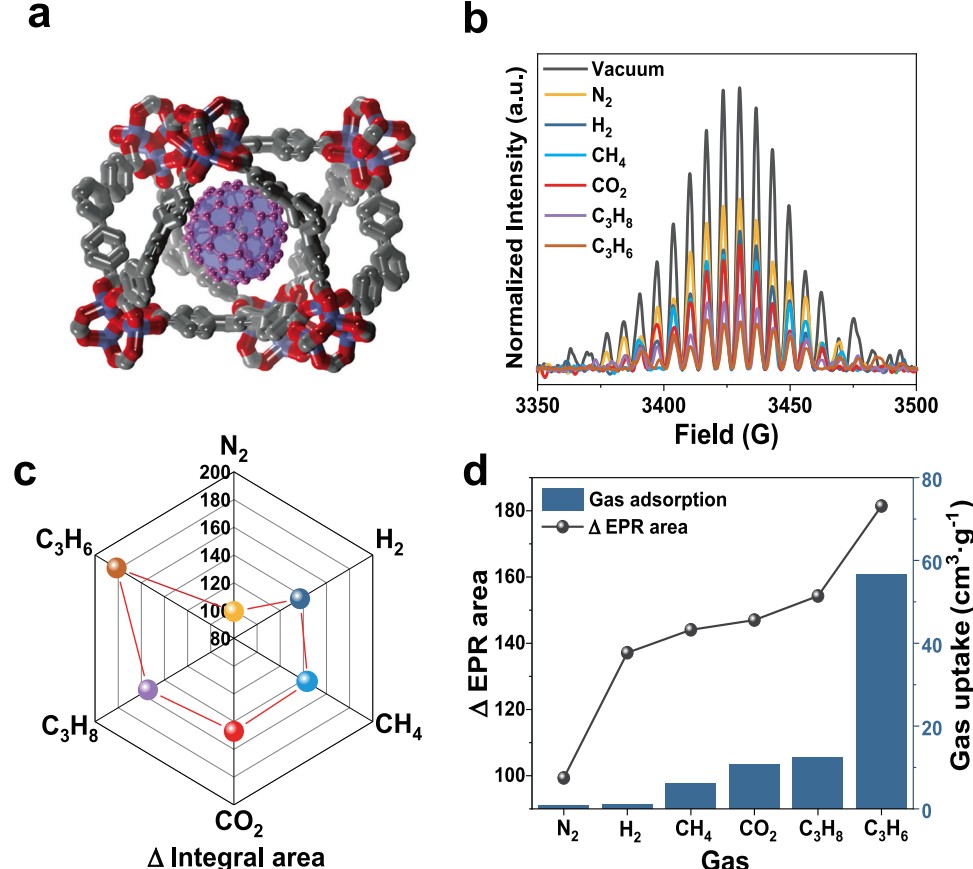

**Fig. 6 | Probing the gas adsorption process of MOF-177 with the molecular spin of Sc₃C₂@C₈₀.** a Schematic unit structure of Sc₃C₂@C₈₀⊂MOF-177. **b** Integrated EPR spectra of Sc₃C₂@C₈₀⊂MOF-177 under vacuum and after adsorbing N₂, H₂, CH₄, CO₂, C₃H₆, and C₃H₈. **c** Difference values between the integrated EPR signal area of Sc₃C₂@C₈₀⊂MOF-177 after adsorbing the target gas and the counterpart of Sc₃C₂@C₈₀⊂MOF-177 under vacuum conditions. **d** Measured adsorption amounts of different gases for MOF-177 using adsorption isotherms at 1 bar (histogram), and the changed difference values for the integrated EPR signal areas of Sc₃C₂@C₈₀⊂MOF-177 before and after adsorbing different gases at 1 bar (point plot). Source data are provided as a Source Data file.

quantifies the gas adsorption of all pores in porous organic frameworks, whereas EPR measurements only reflect gas adsorption near Sc₃C₂@C₈₀ in the nanoscale channel of porous organic frameworks. Therefore, the Sc₃C₂@C₈₀ nano spin sensor embedded in porous organic frameworks can probe gas adsorption with high sensitivity.

Another advantage of this nano spin sensor is that the required amounts of porous organic frameworks are small. In this study, a milligram mass of porous organic frameworks (ca. 1 mg) was required for adsorption evaluation with this spin sensor of metallofullerene; however, traditional adsorption isotherm measurements often require significantly more sample amounts (ca. 100 mg). The minimal amount of sample required indicates that this nano spin sensor has promising applications in the field of trace sensing, wherein a small amount of porous material can facilitate numerous gas adsorption evaluations.

This study revealed the significant interaction between the metallofullerene spin and gas molecules in nanospace. Thus, the molecular spins of metallofullerene can be employed to probe gas adsorption inside the pores of porous organic frameworks based on their quantum characteristics, demonstrating that metallofullerene-based molecular spins are applicable for quantum sensing applications.

## Methods
### Preparation of Sc₃C₂@C₈₀
Metallofullerene Sc₃C₂@C₈₀ was synthesized based on the Kräschmer–Huffman arc-discharge method using an arc-discharge generator. All reagents were obtained commercially and used without further

purification unless otherwise specified. The products were extracted from carbon ash via ultrasound in an ortho-dichlorobenzene (o-DCB) solvent. The extracts were separated through multistage high-performance liquid chromatography with a Buckyprep column (20 × 250 mm) and a Buckyprep-M column (20 × 250 mm) using toluene as the eluent. Sc₃C₂@C₈₀ was detected using a matrix-assisted laser desorption/ionization time-of-flight mass spectrometer (AXIMA Assurance, Shimadzu).

### Source of porous organic frameworks
Py-COF was purchased from Chemsoom Co., Ltd. MOF-177 was prepared using a method reported in the literature[56]. Briefly, Zn(NO₃)₂·6H₂O (1.8 g) and 1,3,5-Tri(4-carboxyphenyl)benzene (H₃BTB, 120 mg) were dissolved in N, N-Dimethylformamide (DMF, 36 ml) and sonicated for 1 h. The mixed solution was then sealed in a glass tube and heated at 85 °C for 3500 min. The products were washed three times with DMF and three times with toluene.

### Preparation of Sc₃C₂@C₈₀⊂Py-COF and Sc₃C₂@C₈₀⊂MOF-177
First, Py-COF and MOF-177 samples with large sizes and high degrees of crystallinity were selected for preparation. Then, Py-COF (2 mg) powder was immersed in the toluene solution containing Sc₃C₂@C₈₀ (4 × 10⁻⁵ mol·L⁻¹) for one week to ensure that the Sc₃C₂@C₈₀ molecules completely entered the channels of Py-COF. The resulting brown powder was washed with toluene and vacuum dried at 80 °C for 12 h to obtain the Sc₃C₂@C₈₀⊂Py-COF complex. Sc₃C₂@C₈₀⊂MOF-177 was

prepared according to the method above, and related details are described in Supplementary Note 4.

## XPS measurement

The XPS analysis of $Sc_3C_2@C_{80} \subset Py\text{-}COF$ was performed using the VG Scientific ESCALab220i-XL electron spectrometer with 300 W Al Kα radiation. As is typical, the hydrocarbon C1s line at 284.8 eV from adventitious carbon was used for energy referencing.

## SEM and TEM measurement

The morphologies and microstructures of the samples were observed using a field emission scanning electron microscope (Regulus8100, HITACHI, Japan) operated at a voltage of 10 kV, along with a TEM (JEM-2100F, JEOL, Japan). Element mapping was characterized on a TEM equipped with Oxford detection.

## Electron paramagnetic resonance spectrum measurements

EPR spectra were recorded on an EPR spectrometer (CIQTEK EPR200-Plus) with a continuous-wave X-band frequency (~9.6 GHz). A small quantity of complex (0.86 mg for $Sc_3C_2@C_{80} \subset Py\text{-}COF$ and 5.0 mg for $Sc_3C_2@C_{80} \subset MOF\text{-}177$) was placed at the bottom of a quartz tube. The modulating field strength, frequency, and microwave power were 2 G, 100 kHz, and 2 mW (20 dB), respectively. To probe the gas adsorption of Py-COF using the molecular spin of $Sc_3C_2@C_{80}$, the EPR signals of $Sc_3C_2@C_{80} \subset Py\text{-}COF$ were recorded after adsorbing $N_2$, CO, $CO_2$, $CH_4$, $C_3H_6$, and $C_3H_8$. $Sc_3C_2@C_{80} \subset Py\text{-}COF$ (0.86 mg) was placed in a quartz tube connected to a vacuum pump and gas bag (at 1 bar). $Sc_3C_2@C_{80} \subset Py\text{-}COF$ in the quartz tube was evacuated for 30 min to remove air from the pores before recording EPR signals under vacuum. Then, gases were filled into the quartz tube; after adsorbing for 30 min, the EPR signals were collected. The detailed test procedure is described in Supplementary Note 2.

## PXRD measurements

The solid samples were loaded onto a glass sample holder to record the PXRD spectra on a Rigaku D/max-2500n diffractometer with Cu Kα radiation (λ = 1.541 Å) at 40 kV and 200 mA. The scanning range was from $2\theta = 2°$ to 80° with a scan speed of 2° min$^{-1}$.

## Gas adsorption isotherm measurements

Gas adsorption isotherms were recorded on a BeiShiDe Instrument volumetric adsorption analyzer. Prior to the measurements, 100 mg samples were degassed for over 300 min at 300 °C, and the equilibration time was 120 s. Ultra-high purity $N_2$, CO, $H_2$, $CO_2$, $CH_4$, $C_3H_6$, and $C_3H_8$ were used for measurements. Oil-free vacuum pumps and oil-free pressure regulators were used for measurements to prevent contamination of the samples during the degassing process and isotherm measurement. A liquid $N_2$ bath was used for the adsorption measurements at 77 K. Before measurement, the samples were degassed in vacuum at 150 °C for 300 min. To provide high accuracy and precision in determining P/P$_0$, the saturation pressure P$_0$ was measured throughout the $N_2$ analyses using a dedicated saturation pressure transducer, which allowed us to monitor the vapor pressure at each data point.

## Statistical Information

Statistics were performed either with OriginPro 2021 software. The bar chart data in Fig. 5 represents the mean ± SD (n = 20).

## Data availability

The data that support the findings of this study are available in the online version of this paper and the accompanying Supplementary Information, or available from the corresponding authors on request. Source data are provided with this paper.

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

## Acknowledgements

This work was supported by the National Natural Science Foundation of China (52022098, 51972309) and the Youth Innovation Promotion Association of Chinese Academy of Sciences (Y201910).

## Author contributions

T.W. and C.W. conceived the research and supervised the project. J.Z., W.L. and C.Z. performed the material preparations and EPR measurements. L.L. performed the DFT calculations. T.W. and J.Z. wrote the paper. All authors discussed the results and commented on the manuscript.

## Competing interests

The authors declare no competing interest.
