## [Peer Review File · Nature Communications]

REVIEWERS' COMMENTS

Reviewer #1 (Remarks to the Author):

I am satisfied with the revisions and glad to suggest accepting it.

Reviewer #3 (Remarks to the Author):

The authors have answered all questions.

RESPONSE TO REVIEWERS' COMMENTS

We thank the Reviewers for your thoughtful suggestions and insights, which have enriched the manuscript and produced a better and more balanced account of the research.

Reviewer #1 (Remarks to the Author):

This manuscript demonstrated a metallofullerene-based nano spin sensor for probing the gas adsorption of porous organic frameworks with in-situ and real-time monitoring. What's more, this study disclosed the significant interaction between metallofullerene spin and gas molecules in the nanospace. I think this work has good novelty and deserves to be published. Some issues should be solved as following:

1. All the comparative analysis conclusions of EPR and traditional isothermal adsorption in this paper default that the change of the EPR signal of the same gas under different adsorption amounts is linear. Some literatures or experiment evidences are required to prove this premise.

Response: Thank you for the valuable comment. In this report, the EPR signals of $\text{Sc}_3\text{C}_2@\text{C}_{80}$ within Py-COF or MOF-177 were measured after absorbing N_2 , CO , CH_4 , CO_2 , C_3H_6 , and C_3H_8 . Measurement results show that the changes in EPR signals of embedded $\text{Sc}_3\text{C}_2@\text{C}_{80}$ are associated with the gas adsorption performances of Py-COF or MOF-177. The changed EPR signals of embedded $\text{Sc}_3\text{C}_2@\text{C}_{80}$ after adsorbing different gases can be ascribed to the absorbed layer of gas on frameworks, which can influence the spin-lattice interactions and, consequently, change the EPR signals. In revision, the relationship between EPR signals and gas adsorption is further discussed as follows. Relevant literature references have been added.

“Generally, when gas molecules are adsorbed onto COF or MOF materials, these molecules adhere to the pore frameworks and form gas layers on the frameworks⁴³⁻⁴⁵. These gas layers then restrict the motion of embedded $\text{Sc}_3\text{C}_2@\text{C}_{80}$, intensify spin-lattice interactions, and consequently weaken EPR signals.” (p. 5, line 89–92)

We further added: “In this scenario, more gas molecules would enhance the viscosity of these gas layers and influence the rotational relaxation time of $\text{Sc}_3\text{C}_2@\text{C}_{80}$ molecules. The relationship between the viscosity η and rotational correlation time τ_c in the spin system can be expressed as⁴⁷

$$\tau_c = \frac{4\pi r^3 \eta}{3kT} \quad (1)$$

where r is the dynamic radius of a spin molecule. The spin-lattice relaxation time (T_1) is expressed as follows⁴⁸:

$$\frac{1}{T_1} = \frac{2\pi g \beta_e}{h} (B_x^2 + B_y^2) \frac{\tau_c}{1 + \tau_c^2 \omega_s^2} \quad (2)$$

where $B_x^2 + B_y^2$ denote the square amplitudes of fluctuating fields along the x- and y-directions. According to Equations (1) and (2), a larger viscosity could result in smaller T_1 . Furthermore, the double-integrated intensity (DIN) can be expressed by the Arrhenius equation⁴⁹:

$$\text{DIN}(T) = I_0 \exp\left(\frac{E_a}{kT}\right) \quad (3)$$

where I_0 is a fitting parameter, and E_a is the activation energy required for dissociation of the paramagnetic spin cluster. Combining T_1 with Equation (3) $1/T_1 \propto \exp(-E_a/kT)$ ⁵⁰ reveals that the integrated EPR intensity would decrease with a decrease in T_1 . Therefore, a

corresponding relationship exists between gas adsorption and EPR signals.

(p. 8, line 148–162)

It should be noted that the quantitative relation between EPR parameter and gas adsorption capacity need to be further evaluated by setting up the rational models.

2. In separate adsorption test of Py-COF and MOF-177, combined with the traditional isothermal adsorption measurement results, the larger the adsorption capacity corresponds to the larger Δ EPR. However, comparison between Py-COF and MOF-177 shows that in Py-COF, the traditional gas isothermal adsorption result of C_3H_8 is 130, and the Δ EPR of the probe test is 45. In MOF-177, for C_3H_8 , the traditional gas isothermal adsorption result is less than 20, while the Δ EPR of the probe test is 150. Please explain this phenomenon.

Response: Thank you for the valuable comments. In revision, we added the following under the Discussion to address this in the manuscript:

“The changed difference values for the integrated EPR signal areas (Δ EPR) of $Sc_3C_2@C_{80}$ entrapped within porous materials before and after gas adsorption are dependent on the amount of entrapped $Sc_3C_2@C_{80}$, as well as the interaction between $Sc_3C_2@C_{80}$ and nanopores. A larger amount of embedded $Sc_3C_2@C_{80}$ would result in a larger Δ EPR; however, the interactions between $Sc_3C_2@C_{80}$ and nanopores would influence the EPR signal and Δ EPR. Therefore, a comparative analysis of Δ EPR could be employed for one type of complex. For example, the amount of $Sc_3C_2@C_{80}$ in Py-COF is low (mass fraction: 1.5%); hence, the integrated EPR signal area and Δ EPR are relatively small compared to those of $Sc_3C_2@C_{80}@MOF-177$, which has a higher amount of $Sc_3C_2@C_{80}$ in MOF-177 (mass fraction: 3.78%). In addition, the different interactions between $Sc_3C_2@C_{80}$ and nanopores of Py-COF and MOF-177 would influence the initial EPR signals. Therefore, when the proposed nano spin sensor is employed to probe the gas adsorption performance, Δ EPR after adsorbing one gas should be compared with the counterpart under vacuum conditions for one type of complex.” (p. 20, line 326–337)

3. As mentioned in the paper, compared with the adsorption of N_2 , the rotation of the molecular probe after the adsorption of C_3H_6 is more severely restricted, so the EPR of different gases under the same adsorption amount may be different. Therefore, EPR of different gases under the same adsorption capacity should be explored first, and different gases should be normalized when comparing different adsorption capacities of different gases.

Response: Thank you for the valuable comments. In revision, we compared the EPR signals of different gases at the same adsorption capacity and found that the EPR variations are similar, as shown in Supplementary Fig. 9. These results show that the EPR signals of $Sc_3C_2@C_{80}$ within porous materials mainly depend on the amount of gas around the spin probe for the studied gases. In revision, we have added the above discussions to the manuscript (p. 14 line 242–246).

4. Without proving that EPR changes are linearly related to gas adsorption capacity, the explanation that the adsorption test results of the spin are different from the traditional isothermal adsorption results is also not credible. Therefore, it is necessary to explore whether the change of the EPR signal is linear with gas adsorption.

Response: Thank you for the valuable comment. In revision, the relationship between EPR signals and gas adsorption is further discussed as described in question 1. Relevant literature sources have been added.

5. For “the EPR signals demonstrated more significant differences”, please give a more detailed explanation, because from Supplementary Fig. 5, it seems that the results of traditional isothermal gas adsorption are more obvious.

Response: Thank you for the valuable comment. In this study, the double integrated EPR intensity was employed to evaluate the EPR signal changes after gas adsorption. The integrated intensity can reflect the overall changes and ensure accuracy, but the variable quantity of this parameter is relatively low due to its statistical average characteristic. In revision, this statement of “the EPR signals demonstrated more significant differences” has been removed (originally on p. 12).

6. "22 hyperfine coupling splitting peaks of $\text{Sc}_3\text{C}_2@\text{C}_{80}$ in solution" mentioned in the article have no relevant evidence. Please supplement.

Response: Thank you for the valuable comment. In revision, the EPR spectrum of $\text{Sc}_3\text{C}_2@\text{C}_{80}$ in CS_2 solution has been added, and it can be observed that the $\text{Sc}_3\text{C}_2@\text{C}_{80}$ has 22 splitting peaks due to the couplings between spin and three equivalent Sc nuclei, as shown in Supplementary Fig. 13 (cited on p. 7, line 129-131).

Supplementary Fig. 13 EPR spectrum of $\text{Sc}_3\text{C}_2@\text{C}_{80}$ in CS_2 solution.

7. Please provide SEM or TEM images of Py-COF and $\text{Sc}_3\text{C}_2@\text{C}_{80}\text{Py-COF}$ to characterize whether the morphology and size of the materials are uniformly, and whether the morphology of the material changes after the composite.

Response: Thank you for the valuable comment. In revision, the SEM images of Py-COF and $\text{Sc}_3\text{C}_2@\text{C}_{80}\text{Py-COF}$ have been added; please see Supplementary Fig. 5. From SEM images of Py-COF and $\text{Sc}_3\text{C}_2@\text{C}_{80}\text{Py-COF}$, it can be observed that the material morphology and size does not change after composition. The following was added in revision:

“Scanning electron microscopy (SEM) images of Py-COF and $\text{Sc}_3\text{C}_2@\text{C}_{80}\text{Py-COF}$ revealed that the composite material morphology and size did not change after preparation

(Supplementary Fig. 5). The transmission electron microscopy (TEM) element mapping images and energy dispersive spectroscopy results revealed that $\text{Sc}_3\text{C}_2@\text{C}_{80}$ was uniformly distributed inside Py-COF (Supplementary Fig. 6).” (p. 6, line 121–125)

8. There is no change in the PXRD diagram of Py-COF after embedding $\text{Sc}_3\text{C}_2@\text{C}_{80}$. Please use other tests (XPS, mapping) to prove the successful embedding of the material, rather than simple physical composition.

Response: Thank you for the valuable comment. In revision, the XPS and TEM mapping results have been added; please see Supplementary Fig. 4, Supplementary Fig. 6, and Supplementary Fig. 16. For the complex preparation, after adsorbing $\text{Sc}_3\text{C}_2@\text{C}_{80}$ molecules, the samples were sufficiently washed with toluene to remove the physically adhered $\text{Sc}_3\text{C}_2@\text{C}_{80}$ on the Py-COF surface. We added under Methods:

“The XPS analysis of $\text{Sc}_3\text{C}_2@\text{C}_{80}@\text{Py-COF}$ was performed using the VG Scientific ESCALab220i-XL electron spectrometer with 300 W Al K α radiation. As is typical, the hydrocarbon C1s line at 284.8 eV from adventitious carbon was used for energy referencing.” (p. 23, line 383–385)

In addition, we added:

“The detection of the Sc2p peak in $\text{Sc}_3\text{C}_2@\text{C}_{80}@\text{Py-COF}$ via X-ray photoelectron spectroscopy (XPS) indicated that $\text{Sc}_3\text{C}_2@\text{C}_{80}$ molecules were trapped inside the channels of Py-COF, as shown in Supplementary Fig. 4” (p. 6, line 119–121)

The TEM element mapping images revealed that $\text{Sc}_3\text{C}_2@\text{C}_{80}$ were uniformly distributed inside Py-COF; please see Supplementary Fig. 6. In addition, we added:

“The apparent distinct EPR splitting peaks indicate successful embedment and dispersion of $\text{Sc}_3\text{C}_2@\text{C}_{80}$ molecules, as simple physical dispersion could lead to the aggregation of $\text{Sc}_3\text{C}_2@\text{C}_{80}$, which would present a broad single EPR peak.” (p. 7, line 127–129)

Supplementary Fig. 4. XPS spectrum of Sc2p peak of $\text{Sc}_3\text{C}_2@\text{C}_{80}@\text{Py-COF}$. The peak positions of 2p_{3/2} and 2p_{1/2} are 398.95, 402.4 eV, respectively.

Supplementary Fig. 6. TEM images of $\text{Sc}_3\text{C}_2@\text{C}_{80}\text{Py-COF}$ complex. TEM element mapping images of b C, c O, d N and e Sc for $\text{Sc}_3\text{C}_2@\text{C}_{80}\text{Py-COF}$ complex. f Elemental fractions of $\text{Sc}_3\text{C}_2@\text{C}_{80}\text{Py-COF}$ from EDS spectrum.

Supplementary Fig. 16. TEM images of $\text{Sc}_3\text{C}_2@\text{C}_{80}\text{MOF-177}$ complex. TEM element mapping images of b C, c Zn, d O and e Sc for $\text{Sc}_3\text{C}_2@\text{C}_{80}\text{MOF-177}$ complex. f Elemental fractions of $\text{Sc}_3\text{C}_2@\text{C}_{80}\text{MOF-177}$ from EDS spectrum.

9. Please provide the thermogravimetric analysis of the material to ensure that the material is not damaged when degassing at 300 °C.

Response: Thank you for the valuable comment. In revision, the thermogravimetric analysis

data of Py-COF has been added; please see Supplementary Fig. 1 (cited on p. 6, line 100). This result demonstrates that the Py-COF material is still stable at 300 °C.

Supplementary Fig. 1 TGA curve for Py-COF powder.

10. What are the advantages of $\text{Sc}_3\text{C}_2@\text{C}_{80}$ compared with other spin-active metallofullerene materials?

Response: Thank you for the valuable question. First, $\text{Sc}_3\text{C}_2@\text{C}_{80}$ has a highly stable electron spin protected by the carbon cage, and this stability is the basis of its application. Some other spin-active metallofullerenes, such as $\text{Sc}@\text{C}_{82}^{1,2}$, are not very stable as their electron spins are delocalized on the carbon cage. Second, $\text{Sc}_3\text{C}_2@\text{C}_{80}$ has a highly sensitive electron spin as disclosed by our previous studies. Therefore, these advantages of $\text{Sc}_3\text{C}_2@\text{C}_{80}$ can well be used to detect the gas adsorption in porous materials.

In revision, we added the following to the manuscript:

“ $\text{Sc}_3\text{C}_2@\text{C}_{80}$ has a highly stable electron spin protected by a carbon cage, and this stability forms the basis of its application. Moreover, $\text{Sc}_3\text{C}_2@\text{C}_{80}$ has a highly sensitive electron spin, as reported in our previous studies. These features make $\text{Sc}_3\text{C}_2@\text{C}_{80}$ suitable for gas adsorption detection within porous materials.” (p. 3, line 58–61)

11. Some important literatures related MOF sensing should be cited, such as *Angew. Chem. Int. Ed.* 2022, 61, e202212797; *Nati. Sci. Rev.* 2022, 9, nwac143; *Angew. Chem. Int. Ed.* 2021, 60, 25758-25761

Response: Thank you for the valuable comment. In revision, we have cited these important relevant literatures relating to MOF sensing.

“Porous organic frameworks are emerging crystalline porous materials with broad applications in gas adsorption and separation²⁵⁻²⁸, gas storage^{29,30}, sensing³¹⁻³⁶, and catalysis³⁷⁻³⁹.” (p. 3, line 45-46)

Reviewer #2 (Remarks to the Author):

The author reported an embedded nano spin sensor for in-situ probing of the gas adsorption inside porous organic frameworks. It is very interesting, but I have some questions.

1. How to use EPR spectra to reflect on-site and real-time monitoring? I think visual imaging can be used on-site and real-time monitoring.

Response: Thank you for the valuable comments. The $\text{Sc}_3\text{C}_2@\text{C}_{80}$ spin sensor embedded in porous organic frameworks can probe the gas adsorption and desorption processes with in-situ and real-time monitoring. To illustrate this concept, we designed a group of EPR experiments base on $\text{Sc}_3\text{C}_2@\text{C}_{80}\text{Py-COF}$; N_2 and C_3H_6 gases were selected as detection objects, as shown in Fig. 4. After filling N_2 , the $\text{Sc}_3\text{C}_2@\text{C}_{80}\text{Py-COF}$ demonstrated increased integrated EPR signal areas (ΔEPR) compared to those of $\text{Sc}_3\text{C}_2@\text{C}_{80}\text{Py-COF}$ under vacuum conditions. Furthermore, after successive filling with C_3H_6 , the $\text{Sc}_3\text{C}_2@\text{C}_{80}\text{Py-COF}$ continued to show increased integrated EPR signal areas. Thus, the $\text{Sc}_3\text{C}_2@\text{C}_{80}$ spin sensor embedded in porous organic frameworks can be used to monitor the gas leakage of C_3H_6 with on-site and real-time monitoring in certain conditions.

Thank you for the valuable suggestion about the visual imaging. We think that visual imaging would be realized by analyzing the EPR signals through digital signal analysis software.

2. How to distinguish gas selectivity of N_2 , CO , CH_4 , CO_2 , C_3H_6 , and C_3H_8 ? If there are other gases, how to distinguish the interference of mixed gases?

Response: Thank you for the valuable question. In revision, to address this question we added: “In addition, we compared the EPR signals of different gases at the same adsorption capacity and discovered that the integrated EPR intensity variations were similar, as shown in Supplementary Fig. 9. This indicates that the EPR signals of $\text{Sc}_3\text{C}_2@\text{C}_{80}$ entrapped within porous materials mainly depend on the gas amounts around the spin probe. Therefore, this nano spin sensor is suitable for probing and evaluating the gas adsorption performance of porous materials.” (p. 12, line 212–216)

It should be noted that this nano spin sensor does not have good selectivity when using the parameter of integrated EPR intensity. Thus, other EPR methods, such as pulsed EPR technique, maybe able to distinguish gas selectivity.

3. How to reflect high sensitivity? What is the data of the sensitivity in the study? I suggest that it should be compared with other relevant EPR adsorption methods.

Response: Thank you for the valuable questions. For this nano spin sensor, detecting gas adsorption at low pressure is a significant means of reflecting the sensitivity. To investigate the sensitivity of this spin probe at low pressure, we measured the EPR signals at 0.2 bar and 0.6 bar for the gases N_2 , CO , CH_4 , and CO_2 , as shown in Fig. 3a and 3b. After adsorbing N_2 , CO , CH_4 , and CO_2 at 0.2 bar and 0.6 bar, the EPR signal intensity of $\text{Sc}_3\text{C}_2@\text{C}_{80}\text{Py-COF}$ obviously decreased compared to that under vacuum conditions. The trends of the changed difference values for the integrated EPR signals of $\text{Sc}_3\text{C}_2@\text{C}_{80}\text{Py-COF}$ before and after adsorbing different gases at 0.2 bar and 0.6 bar are shown in Fig. 3c. It can be observed that

the difference values for the integrated EPR signals are significant under low pressure, revealing the high sensitivity of this molecular spin probe based on metallofullerene $\text{Sc}_3\text{C}_2@\text{C}_{80}$. In revision, we measured the EPR signals after adsorbing some gases under lower pressure, as shown in Supplementary Fig. 9. The revised section reads:

“Moreover, the EPR signals after gas adsorption under lower pressures were recorded, as shown in Supplementary Fig. 9. These EPR tests were performed under the following conditions: 0.12 bar CO_2 , 0.03 bar C_3H_8 , and 0.04 bar C_3H_6 . $\text{Sc}_3\text{C}_2@\text{C}_{80}@\text{Py-COF}$ was able to sense gas adsorption at extremely low pressures, revealing the high sensitivity of this molecular spin probe based on metallofullerene $\text{Sc}_3\text{C}_2@\text{C}_{80}$.” (p. 12, line 212–216)

Relevant EPR methods for probing the gas adsorption are rarely reported. In revision, we discussed other relevant EPR reports in the literature as follows:

“To date, relevant EPR methods to probe gas adsorption have seldom been reported; yet there are few reports on the EPR properties of TEMPO radicals within MOF materials. For example, for a TEMPO radical and ZIF-8 complex, the adsorption of different gases at atmospheric pressure was found to change the rotational correlation time of the TEMPO radical⁵². In addition, the TEMPO radical and ZIF-8 complex were involved in investigations on high-pressure induced amorphization. We consider this molecular spin probe based on metallofullerene $\text{Sc}_3\text{C}_2@\text{C}_{80}$ for the gas adsorption measurement of porous organic frameworks to be closer to sensing application⁵³.” (p. 12–13, line 217–223)

4. In the part of Preparation of $\text{Sc}_3\text{C}_2@\text{C}_{80}@\text{Py-COF}$, how to ensure that the $\text{Sc}_3\text{C}_2@\text{C}_{80}$ molecules fully entered the channels of Py-COF?

Response: Thank you for the valuable question. Py-COF was selected due to its pore size (approximately 1.38 nm), which is very suitable to accommodate $\text{Sc}_3\text{C}_2@\text{C}_{80}$ (approximately 0.8 nm in size). For the complex preparation, after absorbing $\text{Sc}_3\text{C}_2@\text{C}_{80}$ molecules, the samples were sufficiently washed with toluene to remove the physically adhered $\text{Sc}_3\text{C}_2@\text{C}_{80}$ on the Py-COF surface. The TEM element mapping images reveal that $\text{Sc}_3\text{C}_2@\text{C}_{80}$ was uniformly distributed inside Py-COF; please see Supplementary Fig. 6. In addition, we added: “The apparent distinct EPR splitting peaks indicate successful embedment and dispersion of $\text{Sc}_3\text{C}_2@\text{C}_{80}$ molecules, as simple physical dispersion could lead to the aggregation of $\text{Sc}_3\text{C}_2@\text{C}_{80}$, which would present a broad single EPR peak.” (p. 7, line 127–129)

5. How did $\text{Sc}_3\text{C}_2@\text{C}_{80}$ and Py-COF combine into $\text{Sc}_3\text{C}_2@\text{C}_{80}@\text{Py-COF}$? Is there any chemical bond or force between both of them? Can all ball-like molecules with sensitive electron spin enter the pores of Py-COF?

Response: Thank you for the valuable questions. The aromatic fullerene shell of metallofullerenes endows them with lipophilicity, making them Py-COF nanopore-accessible. That is because Py-COF is synthesized from bifunctional monomeric 1,6-bis(4-formylphenyl)-3,8-bis(4-aminophenyl)pyrene through intermolecular self-condensation, forming 2D pyrene-based imine COF (see p. 5). Therefore, aromatic pyrene-based frameworks of Py-COF can well accommodate $\text{Sc}_3\text{C}_2@\text{C}_{80}$ through host–guest interactions which include van der Waals forces and certain π – π interactions. Thus, fullerenes and endohedral fullerenes with suitable

size can enter the pores of Py-COF (about 1.38 nm). In revision, the following addition was made:

“The theoretical pore size of Py-COF is 1.38 nm; thus, a pore of Py-COF can effectively accommodate a metallofullerene $\text{Sc}_3\text{C}_2@\text{C}_{80}$ unit (approximately 0.8 nm in size) through host-guest interactions, including van der Waals forces and certain π - π interactions.” (p. 6, line 105–106)

Reviewer #3 (Remarks to the Author):

This manuscript reports an embedded nano spin sensor for in-situ probing of the gas adsorption inside porous organic frameworks. A series of methods were employed to characterize the obtained materials. The experimental results are clearly explained. However, the main hypothesis of probing the gas adsorption of porous organic frameworks is not convincingly demonstrated in the present manuscript and further in-deep experiment and analysis is required to build the relationship between pristine COF and metallofullerene embedded COF. A major revision is required.

1. Just as authors said, “the commonly used adsorption isotherm test measures the gas adsorption of all pores of Py-COF, whereas the EPR measurement only reflects the gas adsorption near $\text{Sc}_3\text{C}_2@\text{C}_{80}$ in the nanoscale channel of Py-COF.” The essential issue is “The pore size of Py-COF shows a theoretical pore size of 1.38 nm”, whereas “ $\text{Sc}_3\text{C}_2@\text{C}_{80}$ (approximately 0.8 nm size)”, which means $\text{Sc}_3\text{C}_2@\text{C}_{80}$ must influence the entrance and property of adsorbing gas molecules near it. Because $\text{Sc}_3\text{C}_2@\text{C}_{80}$ is large enough compared with the pore size, the gas adsorption property of $\text{Sc}_3\text{C}_2@\text{C}_{80}\text{-Py-COF}$ is totally different with Py-COF, and from the principle of EPR measurement, this difference is unrelated with the loading quantity of $\text{Sc}_3\text{C}_2@\text{C}_{80}$, which should be reflected to the EPR experiment results. For example, if authors test adsorptions of C4-C6 hydrocarbons, results of $\text{Sc}_3\text{C}_2@\text{C}_{80}\text{-Py-COF}$ and Py-COF may even be opposite, because $\text{Sc}_3\text{C}_2@\text{C}_{80}$ will block the pores of COF and prevent the gas to infiltrate. In this situation, in-deep experiment and analysis is required, to build the relationship between $\text{Sc}_3\text{C}_2@\text{C}_{80}\text{-Py-COF}$ and Py-COF for adsorption of same gas.

Response: Thank you for the valuable comments. The encapsulation of $\text{Sc}_3\text{C}_2@\text{C}_{80}$ inside the channel of Py-COF would not severely block the pores of Py-COF and prevent the gas from infiltrating as the kinetic diameters of C_3H_6 and C_3H_8 are about 0.40 and 0.42 nm respectively³. In addition, the content of $\text{Sc}_3\text{C}_2@\text{C}_{80}$ in Py-COF is low (mass fraction: 1.5%) and $\text{Sc}_3\text{C}_2@\text{C}_{80}$ molecules are highly dispersed inside the pores; therefore, the C_3H_6 and C_3H_8 gases can infiltrate into the pores from multiple directions. In revision, we performed contrast adsorption isotherm experiments of C_3H_6 and C_3H_8 gases for Py-COF and $\text{C}_{60}\text{-Py-COF}$ (mass fraction: 9.8%); see Supplementary Fig. 8, which shows that the adsorption capacity of Py-COF does not change after fullerene encapsulation. These results show that a small amount of fullerenes inside Py-COF would not prevent the gas from infiltrating. We added the following to the manuscript:

“In addition, we performed contrast adsorption isotherm experiments on C_3H_6 and C_3H_8 gases for Py-COF and $\text{C}_{60}\text{-Py-COF}$ (mass fraction: 9.8%), indicating that the adsorption capacity of Py-COF did not change after fullerene encapsulation (Supplementary Fig. 8). Thus, a small amount of fullerene inside Py-COF would not prevent the gas from infiltrating.” (p. 10–11, line 190–193)

Supplementary Fig. 8 Measured adsorption isotherms of Py-COF and C₆₀□Py-COF (mass fraction: 9.8%) for the C₃H₆ and C₃H₈ gases.

2. Other parts need to be modified: for TOC, necessary text description should be added to explain the concept of each part of materials. The Conclusions part is too long, need to simplify. The format of References should be checked more carefully.

Response: Thank you for the valuable comments. In revision, necessary text descriptions have been added to explain the concept of each part of materials for TOC (see below). In addition, the conclusion part has been simplified to avoid repetition, however, as the manuscript is being resubmitted to *Nature Communications*, the Conclusions section is now part of the “Discussion”, so the section has been adapted as necessary (see Discussion; p. 19–21, line 309–359). The format of the References has also been checked carefully and corrected.

References

1. Cantone AL, Buitelaar MR, Smith CG, Anderson D, Jones GAC, Chorley SJ, *et al.* Electronic transport characterization of Sc@C₈₂ single-wall carbon nanotube peapods. *J. Appl. Phys.* 2008, **104**(8).
2. Inakuma M, Shinohara H. Temperature-Dependent EPR Studies on Isolated Scandium Metallofullerenes: Sc@C₈₂(I, II) and Sc@C₈₄. *J. Phys. Chem. B* 2000, **104**(32): 7595-7599.
3. Zhou S, Wei Y, Li L, Duan Y, Hou Q, Zhang L, *et al.* Paralyzed membrane: Current-driven synthesis of a metal-organic framework with sharpened propene/propane separation. *Sci. Adv.* 2018, **4**(10): eaau1393.

REVIEWER COMMENTS

Reviewer #1 (Remarks to the Author):

This manuscript demonstrated a metallofullerene-based nano spin sensor for probing the gas adsorption of porous organic frameworks with in-situ and real-time monitoring. What's more, this study disclosed the significant interaction between metallofullerene spin and gas molecules in the nanospace. I think this work has good novelty and deserves to be published. Some issues should be solved as following:

1. All the comparative analysis conclusions of EPR and traditional isothermal adsorption in this paper default that the change of the EPR signal of the same gas under different adsorption amounts is linear. Some literatures or experiment evidences are required to prove this premise.
2. In separate adsorption test of Py-COF and MOF-177, combined with the traditional isothermal adsorption measurement results, the larger the adsorption capacity corresponds to the larger Δ EPR. However, comparison between Py-COF and MOF-177 shows that in Py-COF, the traditional gas isothermal adsorption result of C₃H₈ is 130, and the Δ EPR of the probe test is 45. In MOF-177, for C₃H₈, the traditional gas isothermal adsorption result is less than 20, while the Δ EPR of the probe test is 150. Please explain this phenomenon.
3. As mentioned in the paper, compared with the adsorption of N₂, the rotation of the molecular probe after the adsorption of C₃H₆ is more severely restricted, so the EPR of different gases under the same adsorption amount may be different. Therefore, EPR of different gases under the same adsorption capacity should be explored first, and different gases should be normalized when comparing different adsorption capacities of different gases.
4. Without proving that EPR changes are linearly related to gas adsorption capacity, the explanation that the adsorption test results of the spin are different from the traditional isothermal adsorption results is also not credible. Therefore, it is necessary to explore whether the change of the EPR signal is linear with gas adsorption.
5. For "the EPR signals demonstrated more significant differences", please give a more detailed explanation, because from Supplementary Fig. 5, it seems that the results of traditional isothermal gas adsorption are more obvious.
6. "22 hyperfine coupling splitting peaks of Sc₃C₂@C₈₀ in solution" mentioned in the article have no relevant evidence. Please supplement.
7. Please provide SEM or TEM images of Py-COF and Sc₃C₂@C₈₀/Py-COF to characterize whether the morphology and size of the materials are uniformly, and whether the morphology of the material changes after the composite.
8. There is no change in the PXRD diagram of Py-COF after embedding Sc₃C₂@C₈₀. Please use other tests (XPS, mapping) to prove the successful embedding of the material, rather than simple physical composition.
9. Please provide the thermogravimetric analysis of the material to ensure that the material is not damaged when degassing at 300 °C.
10. What are the advantages of Sc₃C₂@C₈₀ compared with other spin-active metallofullerene materials?
11. Some important literatures related MOF sensing should be cited, such as Angew. Chem. Int. Ed. 2022, 61, e202212797; Nati. Sci. Rev. 2022, 9, nwac143; Angew. Chem. Int. Ed. 2021, 60, 25758-25761

Reviewer #2 (Remarks to the Author):

The author reported an embedded nano spin sensor for in-situ probing of the gas adsorption inside porous organic frameworks. It is very interesting, but I have some questions.

How to use EPR spectra to reflect on-site and real-time monitoring? I think visual imaging can be used on-site and real-time monitoring.

How to distinguish gas selectivity of N₂, CO, CH₄, CO₂, C₃H₆, and C₃H₈? If there are other gases, how to distinguish the interference of mixed gases?

How to reflect high sensitivity? What is the data of the sensitivity in the study? I suggest that it should be compared with other relevant EPR adsorption methods.

In the part of Preparation of Sc₃C₂@C₈₀-Py-COF, how to ensure that the Sc₃C₂@C₈₀ molecules fully entered the channels of Py-COF?

How did Sc₃C₂@C₈₀ and Py-COF combine into Sc₃C₂@C₈₀-Py-COF? Is there any chemical bond or force between both of them? Can all ball-like molecules with sensitive electron spin enter the pores of Py-COF?

Reviewer #3 (Remarks to the Author):

This manuscript reports an embedded nano spin sensor for in-situ probing of the gas adsorption inside porous organic frameworks. A series of methods were employed to characterize the obtained materials. The experimental results are clearly explained. However, the main hypothesis of probing the gas adsorption of porous organic frameworks is not convincingly demonstrated in the present manuscript and further in-deep experiment and analysis is required to build the relationship between pristine COF and metallofullerene embedded COF. A major revision is required.

Just as authors said, "the commonly used adsorption isotherm test measures the gas adsorption of all pores of Py-COF, whereas the EPR measurement only reflects the gas adsorption near Sc₃C₂@C₈₀ in the nanoscale channel of Py-COF." The essential issue is "The pore size of Py-COF shows a theoretical pore size of 1.38 nm", whereas "Sc₃C₂@C₈₀ (approximately 0.8 nm size)", which means Sc₃C₂@C₈₀ must influence the entrance and property of adsorbing gas molecules near it. Because Sc₃C₂@C₈₀ is large enough compared with the pore size, the gas adsorption property of Sc₃C₂@C₈₀-Py-COF is totally different with Py-COF, and from the principle of EPR measurement, this difference is unrelated with the loading quantity of Sc₃C₂@C₈₀, which should be reflected to the EPR experiment results. For example, if authors test adsorptions of C₄-C₆ hydrocarbons, results of Sc₃C₂@C₈₀-Py-COF and Py-COF may even be opposite, because Sc₃C₂@C₈₀ will block the pores of COF and prevent the gas to infiltrate. In this situation, in-deep experiment and analysis is required, to build the relationship between Sc₃C₂@C₈₀-Py-COF and Py-COF for adsorption of same gas.

Other parts need to be modified: for TOC, necessary text description should be added to explain the concept of each part of materials. The Conclusions part is too long, need to simplify. The format of References should be checked more carefully.